# Serotonin *N*-acetyltransferase SlSNAT2 Positively Regulates Tomato Resistance Against *Ralstonia solanacearum*

**DOI:** 10.3390/ijms26136530

**Published:** 2025-07-07

**Authors:** Yixi Wang, Gengshou Xia, Xinyi Xie, Hao Wang, Lingyun Zheng, Zhijie He, Junxian Ye, Kangtong Xu, Qi Shi, Hui Yang, Yan Zhang

**Affiliations:** Department of Landscape and Horticulture, Ecology College, Lishui University, Lishui 323000, China; yxwangls@163.com (Y.W.); lsxyxgs@126.com (G.X.); 15988591255@163.com (X.X.); 17805841722@163.com (H.W.); 18457818100@163.com (L.Z.); 17857878925@163.com (Z.H.); 15168040653@163.com (J.Y.); 15268767328@163.com (K.X.); 13705894933@163.com (Q.S.); lsxyyh@126.com (H.Y.)

**Keywords:** melatonin, SlSNAT2, resistance to *Ralstonia solanacearum*, tomato, multiple signaling pathway, mechanisms

## Abstract

Bacterial wilt (BW) is a globally serious soil-borne disease in a wide range of plants, caused by diverse strains of *Ralstonia solanacearum*. However, there are few research reports on melatonin regulating plant resistance against *R. solanacearum*. *N*-acetyltransferase SlSNAT2 is a rate-limiting enzyme in plant melatonin synthesis. This study elucidates the mechanisms of SlSNAT2 modulating tomato resistance to BW. *SlSNAT2* was expressed in tomato roots, stems, and leaves and induced upon *R. solanacearum* inoculation. Knocking out *SlSNAT2* significantly decreased the melatonin content in CRISPR/Cas9 mutant *slsnat2*. With *R. solanacearum* inoculation, the morbidity and disease index value of *slsnat2* were significantly higher than those of the tomato wild-type plant Micro-Tom (MT) according to the wilt rate and severity. The chlorophyll levels, photosynthetic rates, and callus deposition quantity in *slsnat2* were notably lower while the reactive oxygen species (ROS) level was considerably higher than those in the MT after inoculation. Additionally, the *SlSNAT2* deficiency depressed the expression of the mitogen-activated protein kinase (MAPK) pathway genes (*SlMPK1*, *SlMKK2*), salicylic acid pathway genes (*SlGluA*, *SlPR-1a*), jasmonic acid pathway gene *SlPin2*, and pathogenesis-related (PR) protein genes (*SlPR-STH2a*, *SlPR-STH2b*, *SlPR-STH2c*, *SlPR-STH2d*). These results revealed SlSNAT2 enhanced the tomato resistance against *R. solanacearum* by orchestrating ROS homeostasis, callose deposition, MAPK signaling, hormone pathways, and PR gene transcripts.

## 1. Introduction

Bacterial wilt (BW) is a destructive soil-borne disease. It is caused by diverse strains of the bacterium *Ralstonia solanacearum* (*R. solanacearum*). *R. solanacearum* can infect over 200 host plant species across 50 families, particularly those in the Solanaceae family [1,2]. *R. solanacearum* invades the plant xylem through the intercellular spaces of roots and wounds. It subsequently reproduces in situ and impedes water absorption and transport. This ultimately leads to plant dehydration and death [3].

Tomato (*Solanum lycopersicum*) is one of the most widely cultivated vegetables globally, valued for its high nutritional and economic significance [4,5,6]. Tomato serves as a host plant for *R. solanacearum*, with BW typically causing significant losses of more than 60% in the tomato fruit yield in Nigeria [7]. During the process of resisting *R. solanacearum*, tomatoes produce reactive oxygen species (ROS), such as hydrogen peroxide (H_2_O_2_). The intensity of the infection correlates positively with the ROS production [8]. Simultaneously, the antioxidant system mitigates the oxidative damage caused by excessive ROS in the tomato cells [9]. Additionally, callose deposition functions as both a chemical and a physical defense mechanism, reinforcing plant cell walls and inhibiting the spread of *R. solanacearum* [10].

Melatonin is a small indoleamine compound widely present in animals and plants [11], and first identified in plants in 1995 [12,13]. Melatonin plays an important role in plant growth and development, including seed germination regulation [14], lateral root formation [15], flowering delay [16], maturation and senescence control [17], photosynthetic efficiency improvement [18], and crop yield enhancement [19]. Furthermore, melatonin protects plants from various abiotic and biotic stresses, including cold, heat, UV, salinity, drought, waterlogging, heavy metals, and pathogen infections [20].

Plant melatonin constitutes an effective defense pathway against various pathogenic bacterial infections [21,22]. Exogenous melatonin enhances *Arabidopsis* resistance against *Pseudomonas syringae* pv. *tomato* DC3000 (*Pst* DC3000) by increasing the cell wall invertase (CWI) activity, strengthening the cell walls, and promoting callose deposition (cellulose, xylose, and galactose) [23,24]. In *Arabidopsis*, the induction of endogenous melatonin after infection with the avirulent pathogen *Pseudomonas syringae* DC3000 (avrRpt2) depends on both the H_2_O_2_ and the nitric oxide (NO) levels. MAPKKK3 and oxidative signal-inducible1(OXl1) kinases activate melatonin-mediated immune responses, triggering downstream mitogen-activated protein kinase (MAPK) cascades, such as MAPK3 and MAPK6, and upregulating the expression of the pathogen resistance-related genes, *PR1*, *PR2*, and *PR5*, to resist bacterial infections [25]. Thus, melatonin induces plant disease resistance mainly through modulating ROS and NO levels, promoting callose deposition and other cell wall defense mechanisms, regulating phytohormone levels and signaling, and reprogramming defense-related gene expression. However, the delicate resistance mechanisms through which melatonin induces resistance to *R. solanacearum* in plants remain poorly understood.

Plants use tryptophan as a substrate to synthesize melatonin through a four-step enzymatic reaction, involving five melatonin synthases, including tryptophan decarboxylase (TDC), tryptamine 5-hydroxylase (T5H), tryptophan hydroxylase (TPH), serotonin *N*-acetyltransferase (SNAT), and caffeic acid *O*-methyltransferase (COMT) or *N*-acetyl serotonin methyltransferase (ASMT) [26,27]. TDC and SNAT are rate-limiting enzymes crucial for melatonin synthesis [26]. SNAT significantly contributes to plant resistance to biotic stress. In *Arabidopsis*, melatonin acts upstream of salicylic acid (SA), resulting in decreased endogenous melatonin and SA levels in *snat* mutant lines. The expression of defense genes such as *PR1*, *ICS1*, and *PDF1* also decreases, reducing the resistance to the pathogen *Pst* DC3000 [28]. Conversely, *SNAT* overexpression in *Arabidopsis* enhances the expression of *PR5*, *PR33*, *WRKY1*, *PDF2.2*, and *MYC2*, increases endogenous melatonin and jasmonic acid (JA) levels, enhances superoxide dismutase (SOD) and peroxidase (POD) activities, and improves resistance to Botrytis cinerea [29]. Overexpression of the grape (*Vitis vinifera*) gene *VvSNAT2* in *Arabidopsis* enhances endogenous melatonin, SA, and the chlorophyll content, enhancing the resistance to powdery mildew [30]. Therefore, SNAT enhances plant biotic resistance by increasing the SA and JA content, boosting SOD and POD activities, and inducing defense gene expression. However, the role of SNAT in tomato biotic stress resistance remains unclear.

Currently, regulated BW resistance genes and signaling pathways in tomatoes have been identified, including transcription factors SlWRKY30 and SlWRKY81, as well as the SA, JA, ethylene (Eth), and MAPK pathways [31,32]. However, the molecular mechanism and regulatory network of melatonin regulating tomato BW resistance remain largely unknown. To address these gaps, we investigated the function of SlSNAT2 in regulating tomato resistance to BW. Furthermore, we also examined the melatonin content, the ROS accumulation, the callose deposition levels, and the expression levels of MAPK, SA, JA, Eth pathways, and pathogenesis-related salt tolerance homolog 2 (*PR-STH2*) genes. Comprehensive analysis of the physiological and molecular responses of tomato seedlings to BW provides valuable insights into the complex mechanisms through which melatonin promotes bacterial resistance in plants.

## 2. Results

### 2.1. Bioinformatics Analysis of SlSNAT2 

Tomato serotonin *N*-acetyltransferase SlSNAT2 encodes a protein of 191 amino acids residues (aa.), containing the ‘Acetyltansf_1’ domain (64~172 aa.), ‘NAT-SF’ domain (99~158 aa.), and ‘RimI’ domain (121~189 aa.) (Figure 1A). We identified SNAT2 homologs from representative organisms, including plants (*Arabidopsis* AtSNAT2, rice (*Oryza sativa*) OsSNAT2, pepper (*Capsicum annuum*) CaSNAT2, potato (*Solanum tuberosum*) StSNAT2, eggplant (*Solanum melongena*) SmSNAT2, tobacco NtSNAT2), mammals (human (*Homo sapiens*) HsSNAT2, mice (*Mus musculus*) MmSNAT2), blue–green algae (*Cyanobacteria* cSNAT), yeast (brewer’s yeast (*Saccharomyces cerevisiae*) scAANTA), fungi (*Fusarium oxysporuma* FoSNAT), and bacteria (*Streptococcus pyogenes* SpNAT) (Appendix A). Amino acid sequence analysis revealed that the three domains of SNAT2 are relatively conserved in plants and blue–green algae (Figure 1B). Phylogenetic analysis showed that SlSNAT2 exhibits the highest homology with potato StSNAT2 among 13 species analyzed. Furthermore, apart from plants, blue–green algae display the closest genetic relationship to SlSNAT2. In contrast, the SNAT2 sequences from mammals, yeast, fungi, and bacteria differ significantly from those in plants (Figure 1C).

### 2.2. Transcriptional Analysis of SlSNAT2 in Tomatoes

Given the critical role of *SNAT2* in plant resistance to biotic stress, we investigated whether *SlSNAT2* is involved in resistance to BW. RT-qPCR analysis revealed that *SlSNAT2* was expressed in the roots, stems, and leaves of wild-type tomato ‘Micro-Tom’ (MT), with the highest transcript accumulation observed in the leaves, followed by the stems, and the lowest expression level detected in the roots (Figure 2A). Following inoculation with *R. solanacearum*, the relative expression of *SlSNAT2* in the roots was gradually induced over a 72 h period (Figure 2B), whereas its expression in the leaves decreased after 24 h of inoculation (Figure 2C). These results suggest that *SlSNAT2* may play a significant role in regulating resistance to BW.

### 2.3. Knocking Out SlSNAT2 in Tomatoes Results in a Significant Decrease in Melatonin Levels

The Crispr/Cas9 system was employed to knock out *SlSNAT2* in tomatoes, resulting in three distinct mutant lines: *slsnat2-Δ1*, *slsnat2-Δ2*, and *slsnat2-Δ3*, each with specific deletions in the *SlSNAT2* exon. Specifically, *slsnat2-Δ1*, *slsnat2-Δ2*, and *slsnat2-Δ3* exhibited deletions of 62 bp, 65 bp, and 6 bp, respectively, in the first half of the *SlSNAT2* gene (Appendix A). Compared to control plants, the expression of *SlSNAT2* was significantly reduced in both the roots and the leaves of the *slsnat2* lines (Figure 3A,B). Furthermore, the melatonin content was markedly decreased in the leaves of the *slsnat2* mutants (Figure 3C). These results indicate that the knockout of *SlSNAT2* in tomatoes leads to a significant reduction in the melatonin levels.

### 2.4. The slsnat2 Mutant Tomatoes Exhibited a Reduced Resistance to Bacterial Wilt

To evaluate the function of *SlSNAT2* in BW resistance, MT and *slsnat2* mutants were inoculated with *R. solanacearum*. The symptoms of the different levels of tomato bacterial wilt disease are shown in Figure 4A. Compared with MT, all three *slsnat2* mutant lines showed clear wilt symptoms one day post-inoculation with *R. solanacearum* (Figure 4B and Appendix A), with a morbidity rate of 70% and an incidence index of 25 (Figure 4C,D). At 4 days post-inoculation (dpi), the leaves of the *slsnat2* mutants began to turn yellow, with a morbidity rate of 80%, an incidence index of 30 (Figure 4B–D), a yellowing rate of 60%, and a yellowing index of 21.67 (Figure 4E,F). At 7 dpi, the yellowing phenotype in slsnat2 was more pronounced, with a morbidity rate of 80%, an incidence index of 30, (Figure 4B–D and Appendix A), a yellowing rate of 90%, and a yellowing index of 43.33 (Figure 4E,F). These results indicated that *SlSNAT2* positively regulates tomato BW resistance and delays leaf yellowing caused by *R. solanacearum* infection. Thus, due to the consistency in *SlSNAT2* expression levels, melatonin content, and phenotype responses after *R. solanacearum* inoculation among the three *slsnat2* mutant lines, we selected the *slsnat2-Δ1* mutant line for subsequent assays.

Chlorophyll serves as a biomarker for early disease diagnosis in plants, with more severe stress typically resulting in a lower chlorophyll content [33]. We measured the chlorophyll content and photosynthetic rate of MT and *slsnat2* leaves at 0 and 10 d post-inoculation with *R. solanacearum*. At 0 dpi, there was no significant differences in chlorophyll a, chlorophyll b, total chlorophyll, or the photosynthetic rate between the MT and *slsnat2* leaves. At 10 dpi, compared to 0 dpi, chlorophyll a and the photosynthetic rate showed no significant changes in the MT leaves but significantly decreased in the *slsnat2* leaves. Chlorophyll b and total chlorophyll were significantly reduced in both the MT and *slsnat2* leaves, with a greater reduction observed in *slsnat2* compared to the MT (Figure 4G–J). These results demonstrate that *SlSNAT2* positively regulates the chlorophyll content and the photosynthetic rate in tomato leaves under *R. solanacearum* stress.

### 2.5. The slsnat2 Mutants Exhibit Enhanced R. solanacearum-Induced Reactive Oxygen Species Accumulation

ROS including single oxygen (^1^O_2_), superoxide anion (·O_2_^−^), and hydrogen peroxide (H_2_O_2_), play a crucial role in regulating plant resistance to biotic stress, including BW stress. Excessive ROS accumulation can cause oxidative damage to plant cells. To investigate the ROS levels in MT and *slsnat2* plants at 0 and 24 h post-inoculation (hpi) with *R. solanacearum*, histochemical staining of H_2_O_2_ and ·O_2_^−^ was performed on tomato leaves using 3,3′-diaminobenzidine (DAB) and nitroblue tetrazolium chloride (NBT), respectively. The presence of more brown or blue spots on the leaves indicates elevated levels of H_2_O_2_ or ·O_2_^–^, respectively. The results showed no significant difference in the ROS levels between the MT and *slsnat2* leaves at 0 hpi (Figure 5A,B). Both the H_2_O_2_ and the ·O_2_^−^ levels increased at 24 hpi in both the MT and the *slsnat2* leaves compared to those at 0 hpi (Figure 5A,B). However, the levels of H_2_O_2_ and ·O_2_^−^ were significantly higher in *slsnat2* leaves than in MT leaves at 24 hpi (Figure 5A,B). These findings suggest that knocking out *SlSNAT2* reduces the tomato resistance to BW by impairing its ability to clear the ROS accumulation.

### 2.6. Suppression of Prominent Callose Accumulation in Root and Leaf Cells by Knocking Out SlSNAT2 

Callose deposition in tomato cells is closely tied to tomato resistance against bacterial infection from *R. solanacearum*. Aniline blue staining was used to analyze the callose deposition levels in the roots and leaves of MT and *slsnat2* mutants at 0 and 24 hpi with *R. solanacearum*. Fluorescent dots indicate callose.

As shown in Figure 6, no fluorescent dots were observed in the roots and leaves without staining. After staining, a limited number of fluorescent dots were observed in both the MT and slsnat2 plants at 0 hpi. At 24 hpi, the number of fluorescent dots in the root and the leaf vein cells significantly increased in both the MT and the *slsnat2* plants. However, the fluorescence intensity was markedly lower in *slsnat2* compared to MT (Figure 6A,B,D,E). Quantitative analysis using Image J revealed that the callose deposition area in the roots and leaves of MT was 2.74-fold and 4.51-fold higher, respectively, than that in *slsnat2* (Figure 6C,F). Thus, knocking out *SlSNAT2* suppresses callose deposition in root and leaf vein cells of *R. solanacearum*-inoculated tomato plants, thereby weakening disease resistance.

### 2.7. Expression Analysis of MAPK Pathway-Related Genes in Roots and Leaves of MT and slsnat2 Plants Within 72 h Post-Inoculation with R. solanacearum

Previous studies have demonstrated that tomato MAPK defense pathway signaling genes, such as *SlMPK1*, *SlMPK2*, *SlMPK3*, and *SlMKK2*, positively regulate tomato resistance to BW [31,34]. In MT roots, the expression of *SlMPK1* was significantly induced at 48 hpi, while the *SlMKK2* expression increased approximately 2-fold by 24 hpi. Other genes showed no significant induction in either MT or *slsnat2* within 72 hpi (Figure 7A,B). These results suggest that *SlMPK1* and *SlMKK2* are key genes in the MAPK pathway mediating tomato root BW resistance via *SlSNAT2*.

In leaves, *SlMPK1*, *SlMPK2*, and *SlMKK2* were significantly induced at 24 hpi in MT, with their expression levels increasing by approximately 6-, 4-, and 10-fold, respectively. In *slsnat2* mutants, *SlMPK1* and *SlMKK2* were significantly induced at 48 hpi, with about a 4-fold increase (Figure 7C,D). These findings indicate that *SlMPK1*, *SlMPK2*, and *SlMKK2* are crucial genes in regulating tomato leaf BW resistance through *SlSNAT2*.

### 2.8. Expression Analysis of Salicylic Acid^−^, Jasmonic Acid, and Ethylene Pathway-Related Defense Genes in Roots and Leaves of MT and slsnat2 Plants Inoculated for 72 h

SA, JA, and Eth pathways play an important role in regulating tomato resistance to BW stress and melatonin biosynthesis [20,35]. To elucidate the relationship between SA, JA, and Eth pathways and SlSNAT2-mediated regulation of tomato BW resistance, we analyzed the expression levels of relevant genes in the roots and leaves of MT and *slsnat2* plants inoculated with *R. solanacearum* for 72 h. Among these genes, *SlGluA* and *SlPR-1a* are associated with the SA pathway, *SlPin2* and *SlLoxA* pertain to the JA pathway, and *SlPR-1b* and *SlOsmotin* are linked to the Eth pathway [36].

The results showed that in the roots, the expression levels of the SA pathway genes *SlGluA* and *SlPR-1a* were significantly induced at 12 and 48 hpi in MT, increasing approximately 8- and 4-fold, respectively, while no significantly induction was observed in *slsnat2* (Figure 8A). For the JA pathway, the *SlPin2* expression notably increased by about 8-fold in MT at 12 hpi but only 2-fold at 24 hpi in *slsnat2*. There was no significant change in the *SlLoxA* expression for either genotype (Figure 8B). Regarding the Eth pathway, the *SlPR-1b* expression was induced approximately 2.5 times at 48 hpi in MT but dramatically increased by 20-fold at 72 hpi in *slsnat2*. Additionally, the *SlOsmotin* expression was induced in both MT and *slsnat2* at 12 hpi, increasing approximately 13-fold in MT compared to 2-fold in *slsnat2*, and at 72 hpi, *SlOsmotin* expression surged to approximately 46 times higher than the baseline values in MT (Figure 8C). These results suggest that both the SA and JA pathways, as well as *SlOsmotin* activation within the Eth pathway, positively regulate the tomato root resistance to BW mediated by *SlSNAT2*.

In leaves, the results showed that the *SlGluA* expression significantly increased in MT at 12 hpi, approximately 6-fold, and at 48 hpi, it rose by approximately 18-fold. In *slsnat2*, *SlGluA* expression was induced about 3-fold at 24 hpi. The expression levels of *SlPR-1a* in MT and *slsnat2* were significantly induced at 24 hpi, increasing approximately 165-fold and 65-fold, respectively (Figure 8D). Regarding the JA pathway, *SlPin2* and *SlLoxA* were significantly induced in MT at 24 and 72 hpi, respectively, with expression levels increasing by approximately 80- and 4-fold, respectively. However, they were not significantly induced in *slsnat2* (Figure 8E). Concerning the Eth pathway, the expression trends of *SlPR-1b* in both MT and *slsnat2* were similar within 72 hpi, with respective increases of approximately 3.4- and 2.8-fold at 48 hpi. Additionally, *SlOsmotin* expression in *slsnat2* was significantly induced at 48 hpi, increasing approximately 28-fold (Figure 8F). These findings indicate that the SA, JA, and Eth pathways are involved in SlSNAT2-mediated regulation of tomato leaf resistance to BW, with the SA and JA pathways responding positively.

### 2.9. Expression Analysis of SlWRKY30 and SlWRKY81 in Roots and Leaves of MT and slsnat2 Plants Within 72 h Post-Inoculation with R. solanacearum

The WRKY transcription factors have been implicated in the positive regulation of the tomato resistance to BW [32]. To investigate whether the WRKY transcription factors are associated with SlSNAT2-mediated regulation of tomato BW resistance, we analyzed the expression levels of *WRKY30* and *WRKY81* in the roots and leaves of the MT and *slsnat2* plants inoculated with *R. solanacearum* for 72 h. The results showed that in the roots, the expression of *SlWRKY30* and *SlWRKY81* in *slsnat2* were induced at 72 hpi and 12 hpi, respectively, with increases of approximately 6- and 3-fold. However, their expressions were not significantly induced in MT (Figure 9A). In the leaves, the expression of *SlWRKY30* and *SlWRKY81* in *slsnat2* was significantly induced after 48 hpi, showing increases of approximately 3.5-fold and 14-fold, respectively (Figure 9B). These findings suggest that WRKY transcription factors play a role in regulating the tomato root and the leaf resistance to BW through *SlSNAT2.*

### 2.10. Expression Analysis of SlPR-STHs in Roots and Leaves of MT and slsnat2 Plants Within 72 h Post-Inoculation with R. solanacearum

The pathogenesis-related proteins SlPR-STH2a, SlPR-STH2b, SlPR-STH2c, and SlPR-STH2d have been implicated in the positive regulation of tomato resistance to BW [32]. To determine whether these pathogenesis-related proteins are associated with SlSNAT2-mediated regulation of tomato BW resistance, we analyzed the expression levels of *SlPR-STH2a*, *SlPR-STH2b*, *SlPR-STH2c*, and *SlPR-STH2d* in the roots and leaves of the MT and *slsnat2* plants inoculated with *R. solanacearum* for 72 h.

In the roots, the expression of *SlPR-STH2a* and *SlPR-STH2c* in MT was induced at 12 hpi, with increases of approximately 35- and 7-fold, respectively. In *slsnat2*, the expression of *SlPR-STH2a* was induced at 24 hpi, increasing by about 20-fold, while no significant induction was observed for *SlPR-STH2c*. Additionally, both *SlPR-STH2b* and *SlPR-STH2d* were significantly induced in *slsnat2* following inoculation, reaching maximum induction factors of approximately 300- and 75-fold, respectively, whereas they were not significantly induced in MT (Figure 10A,B). These results indicate that pathogenesis-related proteins play a role in regulating tomato root resistance to BW through *SlSNAT2,* specifically showing positive responses from both *SlPR-STH2a* and *SlPR-STH2c*.

In the leaves, the expression of both *SlPR-STH2a* and *SlPR-STH2b* was significantly induced in MT at 12 hpi but only at 48 hpi in *slsnat2*. Furthermore, the expression of *SlPR-STH2c* was significantly induced at 48 hpi in *slsnat2*, with an overall increase of approximately 43-fold. *SlPR-STH2d* showed significant induction at 12 hpi in MT and at 48 hpi in *slsnat2* (Figure 10C,D). These results indicated that pathogenesis-related proteins are involved in SlSNAT2-mediated regulation of tomato leaf resistance to BW, with *SlPR-STH2a*, *SlPR-STH2b*, and *SlPR-STH2d* exhibiting positive responses.

## 3. Discussion

Melatonin plays an important role in plant resistance to biotic stresses. Compared to endogenous melatonin, there are extensive reports on the mechanisms of exogenous melatonin in contributing plant responses to fungal, viral, and bacterial pathogens. In tomato fruits, exogenous melatonin enhances resistance to *Botrytis cinerea* by regulating H_2_O_2_ levels and JA signaling pathways [37]. In tomato leaves, exogenous melatonin inhibits cell death, reduces ROS accumulation, promotes callose deposition, and induces the expression of defense-related genes such as *SlWRKY33*, *SlMYC2*, and *SlERF*, thereby strengthening resistance to *B. cinerea* [38]. In cotton (*Gossypium hirsutum*), exogenous melatonin increases the resistance to *Verticillium dahliae* by promoting lignin and gossypol biosynthesis [39]. Furthermore, exogenous melatonin has been shown to enhance plant resistance to other fungal pathogens, including *Penicillium* spp., *Fusarium* spp., and *Alternaria* spp. [40]. Similarly, research revealed that exogenous melatonin enhanced plant resistance to viral pathogen. Chen et al. found that 95% of apple (*Malus pumila*) bud tips infected with *Apple stem grooving virus* (ASGV) became virus-free when cultured in a medium supplemented with 15 μM melatonin [41]. In rice, exogenous melatonin increases endogenous melatonin content and nitric oxide (NO) levels, inducing the expression of defense genes *OsPR1b* and *OsWRKY45*, thus enhancing resistance to *rice stripe virus* (RSV) [42]. In eggplants, exogenous melatonin and SA treatment enhance antioxidant enzymes activity and mitigate oxidative damage caused by *Alfalfa mosaic virus* (AMV), thereby increasing the resistance to AMV [43]. Furthermore, roles of exogenous melatonin have been determined in elevating plant resistance to bacteria. Research indicates that exogenous melatonin enhances the antioxidant system in Chinese cabbage (*Brassica rapa*), reducing damage caused by *Plasmodiophora brassicae* [44]. In *Arabidopsis thaliana*, exogenous melatonin increases NO levels, enhancing the resistance to *Pst* DC3000. Additionally, NO scavengers inhibit the induction of SA-related genes (*EDS1*, *PAD4*, *PR1*, *PR2*, and *PR5*) by exogenous melatonin [24]. Exogenous melatonin also increases the SA and ABA levels while reducing the JA accumulation, further enhancing resistance to *Pseudomonas syringae* in *Arabidopsis* [45]. Notably, the expression of *PR* genes induced by exogenous melatonin is almost completely or partially inhibited in *npr1*, *ein2*, and *mpk6 Arabidopsis* mutants, indicating that melatonin-induced plant defenses are dependent on SA and Eth signaling in various degrees [23].

Unlike animals, melatonin synthesis pathways in plants are not exclusive ones but complicated. It is well established that the plant melatonin synthesis enzymes include TDC, T5H, SNAT, and COMT/ASMT. Among these, SNAT acts as the rate-limiting enzyme and plays a pivotal role in melatonin biosynthesis [46]. In *Arabidopsis*, there are two SNAT homologous proteins, AtSNAT1 (At4g19985) and AtSNAT2 (At1g26220; BT005218; Q9C666.1). The amino acid sequence homology between these two proteins is relatively low (14.07%) (Appendix A), and their functions are not entirely the same [47]. AtSNAT1 regulates endoplasmic reticulum stress tolerance, inhibits lipid and anthocyanin accumulation in seeds, and resists high light and osmotic stress [48,49,50,51]. In contrast, AtSNAT2 regulates the carbon assimilation rates and the flowering time [47,52]. Similar to *Arabidopsis*, tomatoes also possess two SNAT homologs, SlSNAT1 (Solyc10g074910) and SlSNAT2 (Solyc05g010250), which exhibit low amino acid homology (only 23.75%, Appendix A).

Although progress has been made in understanding the function of exogenous melatonin on plant resistance to biotic stresses, the mechanisms underlying endogenous melatonin regulating plant resistance to pathogens remain largely unexplored. Moreover, mechanisms of melatonin mediating plant resistance to *Ralstonia solanacearum* are still vague. In addition, there are limited reports on the roles of tomato SlSNAT1 and SlSNAT2 in bacterial resistance, and no studies have been found on exploring these two proteins mediating plant resistance to *R. solanacearum*.

Our previous experiments demonstrated that SlSNAT2 plays a more significantly positive role than SlSNAT1 in regulating the tomato resistance to *R. solanacearum*. Compared with *slsnat1* mutants, *slsnat2* mutants exhibited a more pronounced wilting phenotype and faster disease progression (unpublished data). These findings indicate that SlSNAT2 plays a greater role in regulating the tomato resistance to BW than SlSNAT1. Therefore, we investigated the mechanisms of SlSNAT2 regulating tomato resistance to BW and found that SlSNAT2 affects the melatonin content, reduces the ROS levels, promotes the callose deposition, and regulates the expression of genes associated with MAPK signaling pathways, plant hormone signaling pathways (SA, JA, Eth), and pathogenesis-related proteins (SlPR-STHs) during the process of tomato resistance to BW. It is evident that the key enzyme in melatonin synthesis, SlSNAT2, participates in multiple regulatory pathways to enhance tomato resistance against *R. solanacearum*. Subsequently, we aim to further elucidate the mechanisms for the key enzymes of melatonin synthesis, SNATs, regulating tomato BW resistance based on the acquisition of *slsnat1 slsnat2* double mutant tomatoes.

Callose is a β-(1,3)-D-glucan widely distributed among multicellular green algae and higher plants [53]. It primarily accumulates at the edges of sieve pores and is also distributed in plant cell plates, sieve plates, plasmodesmata, and vascular bundles located in leaf veins and mesophyll. Upon exposure to biotic stress, the callose rapidly accumulates at these sites to block the sieve pores, thereby enhancing physical defense within the plant cell wall and restricting pathogen spread and infection [54]. Our observations revealed that callose deposition in the roots, veins, and mesophyll of MT tomato was significantly higher than in the *slsnat2* mutant, indicating that SlSNAT2 regulates callose deposition in tomatoes to influence the spread of *R. solanacearum* (Figure 6). However, previous research on SNAT’s role in bacterial disease resistance has shown that *Arabidopsis* SNAT primarily relies on SA signaling pathways for resistance to *Pseudomonas* without involving a callose-related mechanism [28]. Similarly, no such mechanism has been identified in SNAT-mediated resistance to fungal stress or in the biotic stress resistance mechanisms of other melatonin synthesis enzymes (TDC, T5H, ASMT, COMT) [29,55]. Nevertheless, Zhao et al. demonstrated that exogenous melatonin treatment in *Arabidopsis thaliana* increased callose deposition within the cells, thereby enhancing the resistance to *Pst* DC3000 [24]. Consequently, it is worth investigating further whether other melatonin synthesis enzymes also regulate callose accumulation to improve the tomato resistance to BW, to clarify whether this pathway is a specific regulatory mechanism of the key melatonin synthesis enzyme SNAT. Additionally, various mechanisms regulating the callose levels in plants have been identified. Research has indicated that pathogen-induced callose deposition is increased by SA and SA analogs such as benzothiadiazole (BTH) [56,57]. Interestingly, we observed a significant reduction in the expression of the SA pathway-related defense genes *SlGluA* and *SlPR-1a* in the *slsnat2* mutant, suggesting that the SA content in this mutant is likely lower than that in the MT, indicating a potential role for SlSNAT2 in regulating the callose deposition through the SA pathway.

The transcription factor SlWRKY30 has been shown to cooperatively enhance the promoter activity of disease resistance proteins SlPR-STHs together with SlWRKY81, thus increasing the expression levels of *SlPR-STHs* and enhancing tomato resistance to *R. solanacearum* [32]. Compared with the MT, the mRNA expression levels of *SlWRKY30* and *SlWRKY81* were significantly induced in the roots and the leaves of *slsnat2* following inoculation with *R. solanacearum* (Figure 9). However, the expression levels of the *SlPR-STHs* were significantly lower than those in the MT (Figure 10). There could be several reasons for this result. Firstly, there may not be consistency between the mRNA level expressions and the protein level expressions for the WRKY transcription factors. Secondly, the SlWRKY30 and SlWRKY81 cooperatively regulate the tomato resistance to *R. solanacearum* and may form functional complexes, and an excess of monomers or an imbalance in their ratio may disrupt the formation and function of these complexes. Finally, compared with the study by Dang et al., differences exist in the tomato seedling age, the *R. solanacearum* strain, the bacterial concentration, and the quantity, as well as the time point of detection after inoculation in this study [32]. These methodological differences may lead to changes in the mRNA levels of *SlWRKY30* and *SlWRKY81* in the short term after inoculation with *R. solanacearum*.

## 4. Materials and Methods

### 4.1. Bacterial Strain, Plant Materials, and Growth Condition

*R. solanacearum* strain GMI1000 (provided by South China Agricultural University) was used in this study. The GMI1000 strain was cultured on TTC medium (Tryptone 10 g/L, casein 1 g/L, glucose 5 g/L, agar 17 g/L, pH 7.0–7.2; supplemented with 0.005% TTC solution after cooling to 60 °C) at 28 °C for 2 d. The tomato cultivar Micro-Tom (MT) was used as a wild-type plant in this study. The mutant plant *slsnat2* was obtained from MT via the CRISRP/Cas9 method. The coding sequence (CDS) of tomato *SlSNAT2* exclusively contains one exon. We obtained three various mutant lines, *slsnat2-Δ1*, *slsnat2-Δ2*, and *slsnat2-Δ3*, with a distinct deletion in the *SlSNAT2* exon. Mutant *slsnat2-Δ1*, *slsnat2-Δ2*, and *slsnat2-Δ3* exhibited total deletions of 62 bp, 65 bp, and 6 bp in the first half of *SlSNAT2*, respectively (Appendix A). The growth conditions for tomato plants were as previously described [38].

### 4.2. Bioinformatics Analysis

The amino acid sequence of SlSNAT2 (*Solanum lycopersicum*, Solyc05g010250) was downloaded from the Sol Genomics Network database (SGN, https://solgenomics.net, accessed on 16 April 2024). The NCBI website was used to analyze SlSNAT2 protein domains (NCBI Conserved Domain Search, https://www.ncbi.nlm.nih.gov/Structure/cdd/wrpsb.cgi, accessed on 27 April 2024). Using the full-length protein sequence of SlSNAT2 as a query, we retrieved SlSNAT2 homologous protein sequences from 12 representative species via NCBI-Protein BLAST (PBLAST, https://blast.ncbi.nlm.nih.gov, accessed on 27 August 2024), and the accession number of these proteins are listed in Appendix A. Homologous protein sequence alignment was performed using DNAMAN software (version: 9.0.1.116) with default settings. The unrooted phylogenetic tree was constructed using MEGA X software (version:12.0.11), the alignment was conducted using the MUSCLE algorithm, and the tree was constructed using the Maximum Likelihood method with the Jones–Taylor–Thornton (JTT) model and 100 bootstrap replicates [58].

### 4.3. Gene Expression Analysis by Quantitative Real-Time PCR (RT-qPCR)

To determine the tissue-specific expression of *SlSNAT2* in MT plants, roots, stems, and leaves of 7-week-old seedlings were collected. To examine the effect of *SlSNAT2* deficiency on gene expression in tomato plants in response to *R. solanacearum* infection, root, and leaf samples from MT or *slsnat2* mutant plants were collected at 0, 12, 24, 48, and 72 hpi. Total RNA and complementary DNA (cDNA) were extracted as previously described with minor modifications [59]. Briefly, the total RNA was isolated using the Promega RNA extraction kit (Promega, Beijing, China), and cDNA was synthesized using the Vazyme reverse transcription kit (Vazyme, Nanjing, China). RT-qPCR was performed using the CFX96 Touch Real-Time PCR Detection System (Bio-Rad, Hercules, CA, USA) with TB Green Premix Ex Taq Kit (TakaRa, Dalian, China). RT-qPCR primers specific for each gene are listed in Appendix A. The *SlACTIN2* gene was used as the tomato internal control. Means of three replicates were used to calculate the relative transcript levels for each sample using the 2^−ΔCt^ or 2^−ΔΔCt^ method [60]. Each experiment was repeated three times.

### 4.4. Endogenous Melatonin Measurement

Endogenous melatonin was extracted and quantified according to the method described by Xia et al. with minor modifications [38]. [_2_H^6^]-melatonin (M215002, Toronto Research Chemicals Ltd., Toronto, ON, Canada) was used as the internal standard. Leaf samples (0.3 g per sample) from 7-week-old tomato plants were ground under liquid nitrogen and then mixed with 1 mL of ethyl acetate containing 10 ng of the internal standard. The mixture was shaken at 4 °C in the dark overnight. After centrifuged at 12,000 rpm for 10 min, the supernatant was collected, and the precipitate was re-extracted with 1 mL of ethyl acetate for an additional 1~2 h at 4 °C in the dark. Following nitrogen blow-down to complete dryness, the residue was dissolved in 250 μL of 70% methanol, and 200 μL of this solution was transferred to an injection vial for detecting. Liquid chromatography-tandem mass spectrometry (LC-MS/MS) (5500+ QTRAP, AB SCIEX, Framingham, MA, USA) was used to quantify the melatonin content. The SCIEX QTRAP 5500 LC-MS/MS system is equipped with a Kinetex^®^ 1.7 µm C^18^ 100 Å chromatographic column (100 mm × 2.1 mm, P/No. 00D-4475-AN, Phenomenex, Torrance, CA, USA). A mixed solution of water (A) and acetonitrile (B) serves as the mobile phase. The injection volume is set at 5 µL, with a flow rate of 0.3 mL/min. The recovery rate is estimated using [_2_H^6^]-melatonin as the internal standard.

### 4.5. Ralstonia solanacearum Inoculation and Disease Index Calculation

The inoculation method was performed according to Qiu et al. with minor modifications [61]. A single colony of GMI1000 was cultured in liquid TTC medium by shaking at 28 °C for 24 h. After incubation, the inoculum concentration was adjusted to an OD_600_ of 0.6. The 7-week-old seedlings were inoculated by wounding the roots and pouring 50 mL of bacterial solution into the soil. The experimental conditions were 28 °C during 14 h of light and 23 °C during 10 h of darkness. The disease index of the tomato bacterial wilt was classified into five levels: level 0, plant healthy; level 1, 0~1/4 leaves wilting; level 2, 1/4~1/2 leaves wilting; level 3, 1/2~2/3 leaves wilting; level 4, plant death. The disease level assessment and the disease index calculation referred to the methods of Zheng et al. [62].

### 4.6. Statistics of Yellowing Index

The yellowing index of tomato leaves is divided into 7 levels: level 0, 0% leaves yellowing; level 1, 0~10% leaves yellowing; level 2, 11~25% leaves yellowing; level 3, 26~49% leaves yellowing; level 4, 50~74% leaves yellowing; level 5, 75~99% leaves yellowing; level 6, 100% leaves yellowing. The methods for judging the yellowing level and calculating the yellowing index were based on the description provided by Li et al. [63].

### 4.7. Determination of Chlorophyll Content and Net Photosynthetic Rate (Pn)

Leaf samples from 7-week-old tomato plants (MT and *slsnat2*) inoculated with *R. solanacearum* were collected at 0 dpi and 10 dpi. Leaves weighing 0.1 g were soaked in 10 mL of 80% acetone for 48 h in the dark until the leaves turned white. OD values were measured at 663 nm and 645 nm, and the chlorophyll content was calculated using the formula described by Niu et al. [64]. A portable photosynthesis analyzer CIRAS-4 (PP SYSTEMS, Amesbury, MA, USA) was used to measure the net photosynthesis rate. The PARi was 600 μmol·m^−2^·s^−1^, and all other settings were kept at default values.

### 4.8. Histochemical Staining of Reactive Oxygen Species

The roots and leaves of 7-week-old MT and *slsnat2* plants inoculated with *R. solanacearum* for 0 h and 24 h were stained using 3,3′-diaminobenzidine (DAB) and nitroblue tetrazolium (NBT) staining solutions. The DAB and NBT assays were conducted as previously described [38]. Each treatment included ten leaves, and the experiments were repeated three times.

### 4.9. Staining and Quantification of Callose Deposition

Callose staining was performed as previously described with minor modifications [39]. Briefly, detached tomato roots and leaves were collected at 0 hpi and 24 hpi and soaked in a mixed liquid of acetic acid and ethanol (*v*:*v* = 1:3) until the samples turned white. After washing three times with 150 mM K_2_HPO_4_, the fixed leaflets were incubated in a staining solution containing 0.01% aniline blue dissolved in 150 mM K_2_HPO_4_ in a 50 mL brown tube for more than 2 h. The stained roots and leaves were then embedded in 50% glycerol and examined under an inverted fluorescence microscope (DMi8, Leica, Wetzlar, GER). Images were captured, and callose deposits were quantified using Image J software (version: 2.14.0) as described by Mason et al. [65]. For image processing, the image type was set to ‘8-bit’ and inverted. Calibration was set to ‘Uncalibrated OD’, with ‘Set Measurements’ of ‘Area’, ‘Integrated density’, and ‘Limit to threshold’.

### 4.10. The Expression Levels of Genes

The roots, stems, and leaves of MT plants were collected. The root and leaf samples of both the MT and the *slsnat2* mutant plants were collected at 0, 12, 24, 48, and 72 hpi. The expression analysis of *SlSNAT2* and BW resistance-related genes was performed, including the MAPK pathway genes (*SlMPK1*, *SlMPK2*, *SlMPK3,* and *SlMKK2*), the SA signaling pathway genes (*SlPR-1a* and *SlGluA*), the JA signaling pathway genes (*SlPin2* and *SlLoxA*), the Eth signaling pathway genes (*SlPR-1b* and *SlOsmotin*), the transcription factors (*SlWRKY30* and *SlWRKY81*), and the pathogenesis-related protein genes (*SlPR-STH2a*, *SlPR-STH2b*, *SlPR-STH2c*, and *SlPR-STH2d*). Specific primers and gene accession numbers can be found in Appendix A.

## 5. Conclusions

In conclusion, this study demonstrated the key melatonin synthesis enzyme, SlSNAT2, enhancing tomato resistance against *R. solanacearum* through the orchestration of ROS homeostasis, callose deposition, MAPK signaling, hormone pathways, and *PR* gene transcripts. Our research provided a solid foundation for further clarifying the molecular mechanisms of SlSNAT2 and melatonin regulating tomato resistance to *R. solanacearum.* At the same time, we propose that SlSNAT2 is a new gene for tomato genetic improvement of BW resistance.

## Figures and Tables

**Figure 1 ijms-26-06530-f001:**
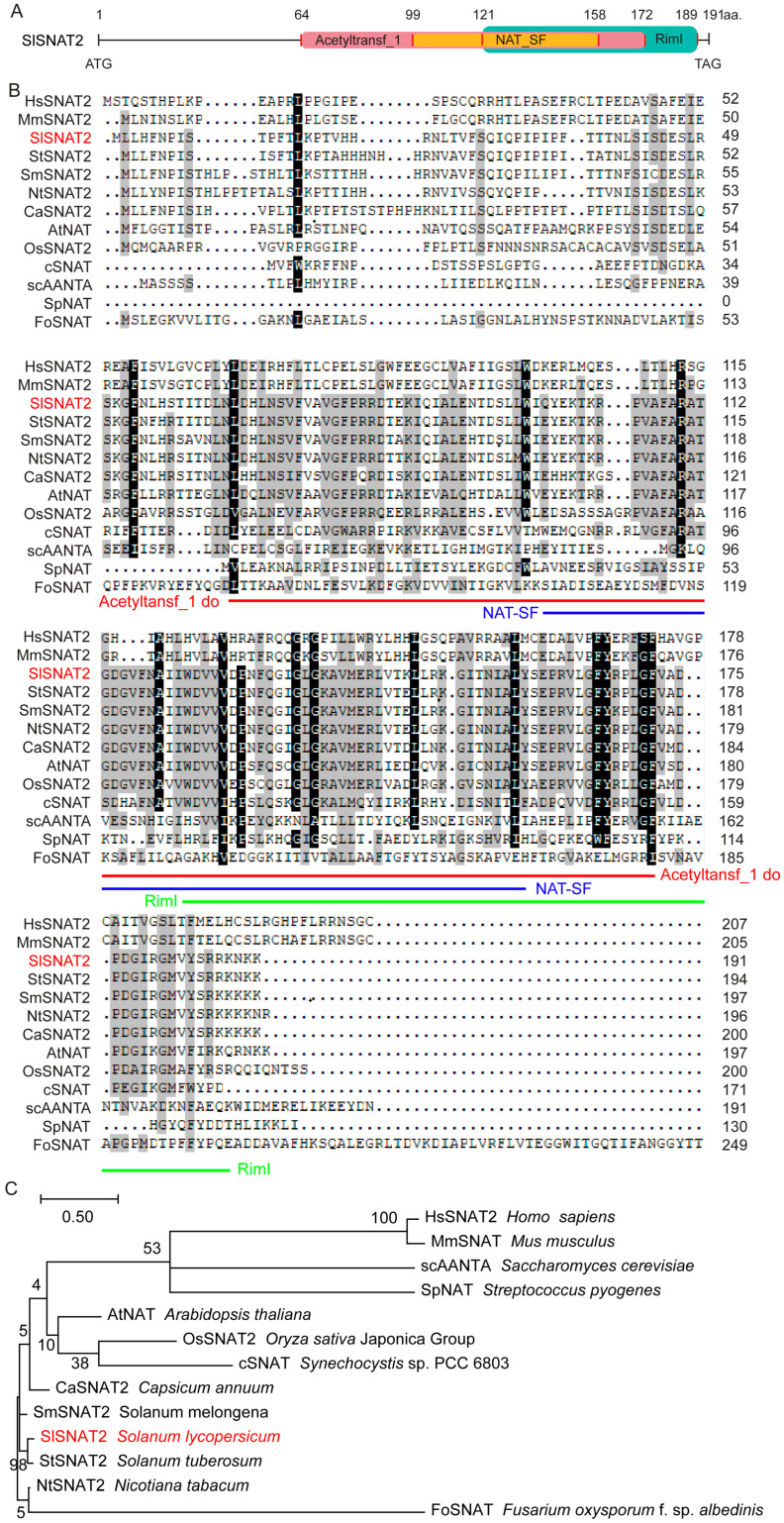
SlSNAT2 conserved domains, amino acid sequences, and phylogenetic analysis. (**A**) Conserved domain analysis of SlSNAT2. Different colored blocks represent different conserved domains. The numbers represent the amino acid sites. (**B**) The amino acid sequences alignment of SlSNAT2 homologous proteins. The amino acid sequence highlighted with red lines corresponds to the ‘Acetyltansf_1’ domain, the yellow-highlighted region corresponds to the ‘NAT-SF’ domain, and the green-highlighted region corresponds to the ‘RimI’ domain. The numbers indicate the positions of the amino acids within the sequence. (**C**) A phylogenetic analysis of SNAT2 among 13 species. MEGA-X was used to construct an evolutionary tree. The number on each branch represents the bootstrap support of the node to its right, out of a maximum value of 100. The accession numbers of the genes are shown in Appendix A.

**Figure 2 ijms-26-06530-f002:**
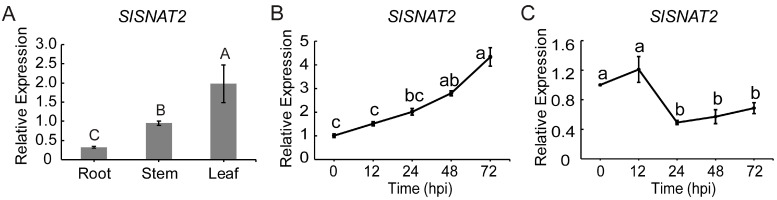
The expression pattern analysis of *SlSNAT2* in tomatoes. (**A**) The relative expression of *SlSNAT2* in tomato tissues. The relative expression was estimated using the 2^−ΔCt^ method. Data are represented as mean ± SEM (*n* = 3). Significant difference analysis was performed using LSD multiple comparisons and represented by letter labeling (*p* < 0.01). The relative expression of *SlSNAT2* in tomato roots (**B**) and leaves (**C**) after inoculation with *R. solanacearum*. The ‘hpi’ stands for the number of hours after inoculation with *R. solanacearum*. The relative expression was estimated using the 2^−ΔΔCt^ method. Data are represented as mean ± SEM (*n* = 3).; Error bars: ±standard deviation with three biological replicates. Significant difference analysis was performed using LSD multiple comparisons and represented by letter labeling (*p* < 0.05). All the experiments were repeated at least three times with similar results.

**Figure 3 ijms-26-06530-f003:**
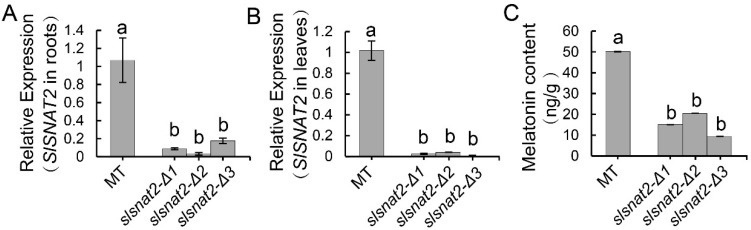
The detection of gene expression and melatonin content in Crispr/Cas9 plants. The relative expression of *SlSNAT2* in roots (**A**) and leaves (**B**) of Micro-Tom (MT) and *slsnat2* mutants. (**C**) The content of melatonin in the leaves of MT and *slsnat2* mutants. The *slsnat2* contains three various mutant lines, and *slsnat2-Δ1*, *slsnat2-Δ2*, and *slsnat2-Δ3*. The relative expression was estimated using the 2^−△△Ct^ method. Data are represented as mean ± SEM (*n* = 3); Error bars: ±standard deviation with three biological replicates. Significant difference analysis was performed using LSD multiple comparisons and represented by letter labeling (*p* < 0.05). All the experiments were repeated at least three times with similar results.

**Figure 4 ijms-26-06530-f004:**
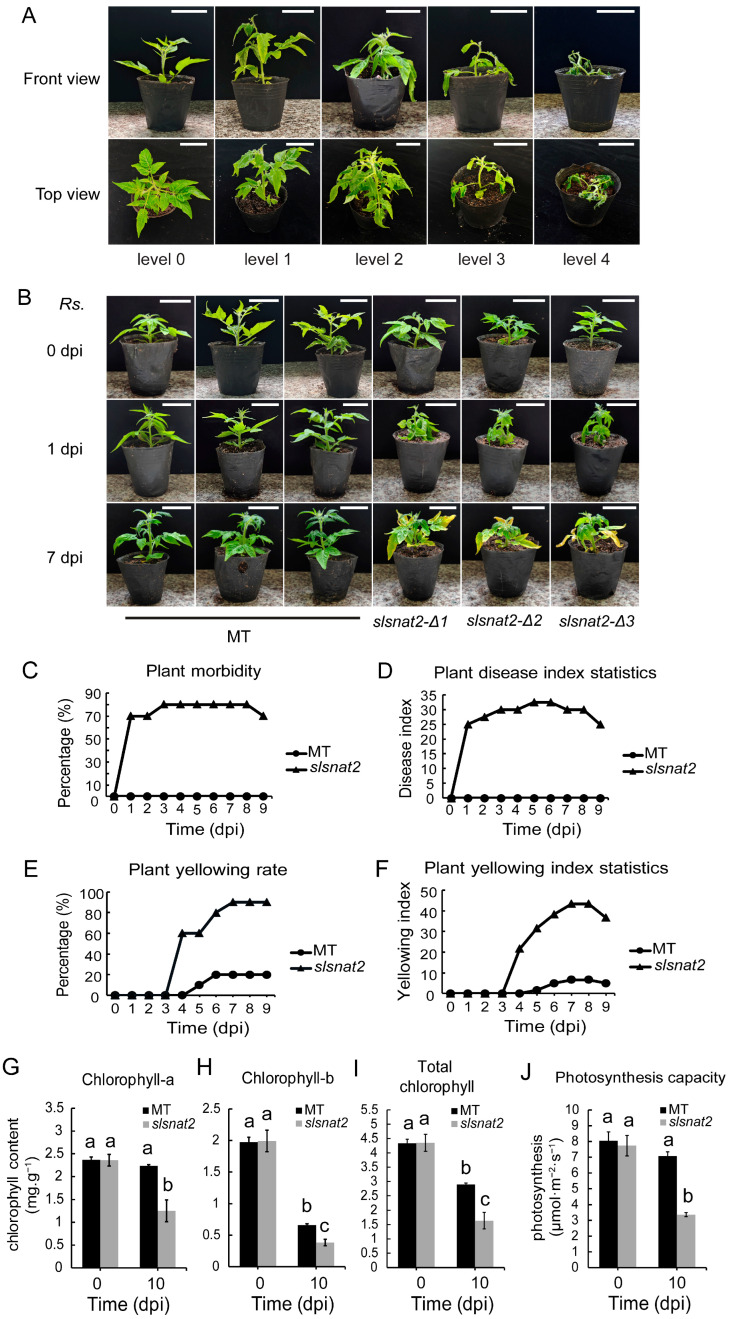
Knocking out *SlSNAT2* reduced the tomato resistance to bacterial wilt. (**A**) The symptoms of different levels of tomato bacterial wilt. Evaluation scale: level 0 = plant healthy, level 1 = 0~1/4 leaf wilting, level 2 = 1/4~1/2 leaf wilting, level 3 = 1/2~2/3 leaf wilting, level 4 = plant death. The white scale bars indicate 5 cm. (**B**) The phenotypes of the Micro-Tom (MT) and slsnat2 plants at 0, 1, and 7 d post-inoculation with *R. solanacearum* in tomatoes. The *Rs.* stands for *Ralstonia solanacearum*. The white scale bars indicate 5 cm. The analysis of morbidity (**C**) and the disease index (**D**), the yellowing rate (**E**) and the yellowing index (**F**) of the MT and *slsnat2* seedlings after inoculation with *R. solanacearum* at over 9 d in tomatoes. The *slsnat2* contains three various mutant lines and *slsnat2-Δ1*, *slsnat2-Δ2*, and *slsnat2-Δ3*. The ‘dpi’ stands for the days post inoculation. Each treatment had at least 10 biological replicates. The analysis of the chlorophyll-a content (**G**), the chlorophyll-b content (**H**), the total chlorophyll content (**I**), and the photosynthetic rate (**J**). Samples (leaves) obtained 10 d after inoculation with *R. solanacearum* were used for analysis. Data are represented as mean ± SEM (*n* = 3). Error bars: ±standard deviation with three biological replicates. Significant difference analysis was performed using LSD multiple comparisons and represented by letter labeling (*p* < 0.05). All the experiments were repeated at least three times with similar results.

**Figure 5 ijms-26-06530-f005:**
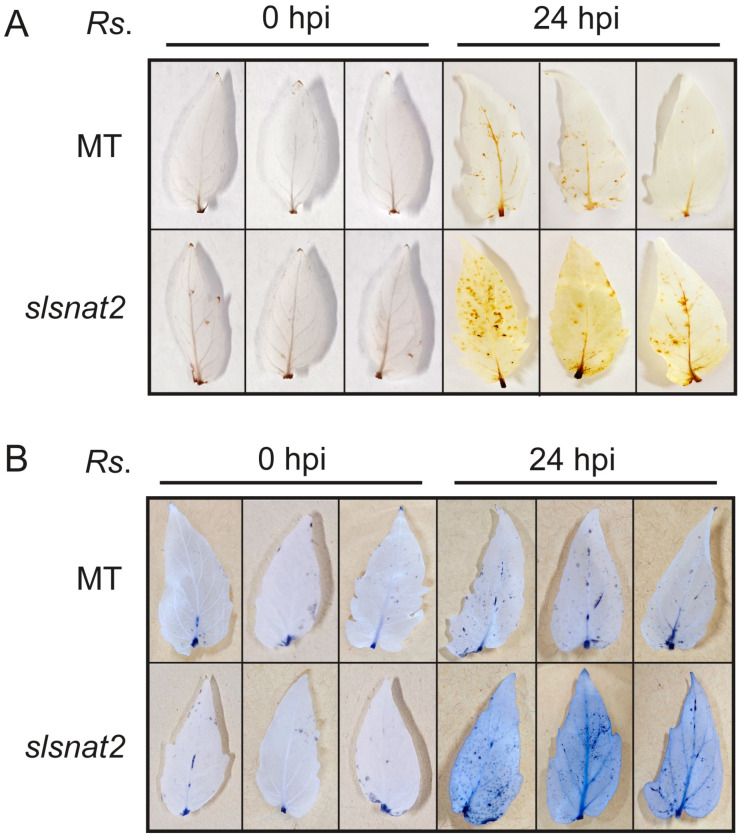
ROS levels in MT and *slsnat2* leaves at 0 and 24 h post inoculation with *R. solanacearum*. (**A**) Detection of hydrogen peroxide (H_2_O_2_) using DAB staining. (**B**) Detection of leaf superoxide (·O_2_^−^) using NBT staining. Inoculated leaves were collected at 0 and 24 h post inoculation of *R. solanacearum* and used for ROS detection using histochemical staining. Each treatment had at least three biological replicates. *Rs.* stands for *R. solanacearum*. All experiments were repeated at least three times with similar results.

**Figure 6 ijms-26-06530-f006:**
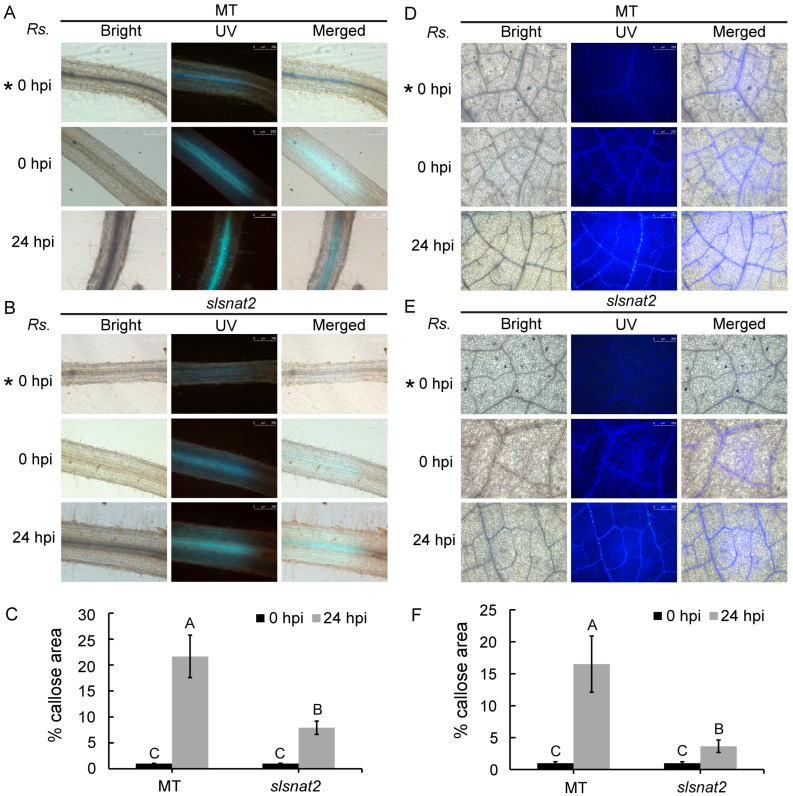
The impact of *SlSNAT2* on the callose deposition induced by *R. solanacearum* in the tomato root cells and the leaf vein cells. The callose deposition in the roots (**A**) and the leaves (**D**) of MT at 24 h post-inoculation with *R. solanacearum*, compared to non-inoculated controls. The Callose deposition in the roots (**B**) and the leaves (**E**) of *slsnat2* at 24 h post-inoculation with *R. solanacearum*, versus non-inoculated controls. *Rs.* represents *R. solanacearum.* The hpi represents the hours post-inoculation. The ‘* 0 hpi’ indicates MT and *slsnat2* roots without aniline blue staining. The white scale bars indicate 250 μm. Each treatment had at least three biological replicates. Quantification of the callose deposition based on the percentage of the stained callose area at 24 h after the *R. solanacearum* infection in the roots (**C**) and the leaves (**F**) of the MT and *slsnat2*. Data are represented as mean ± SEM (*n* = 3); Error bars: ±standard deviation with three biological replicates; Significant difference analysis was performed using LSD multiple comparisons and represented by letter labeling (*p* < 0.05). All the experiments were repeated at least three times with similar results.

**Figure 7 ijms-26-06530-f007:**
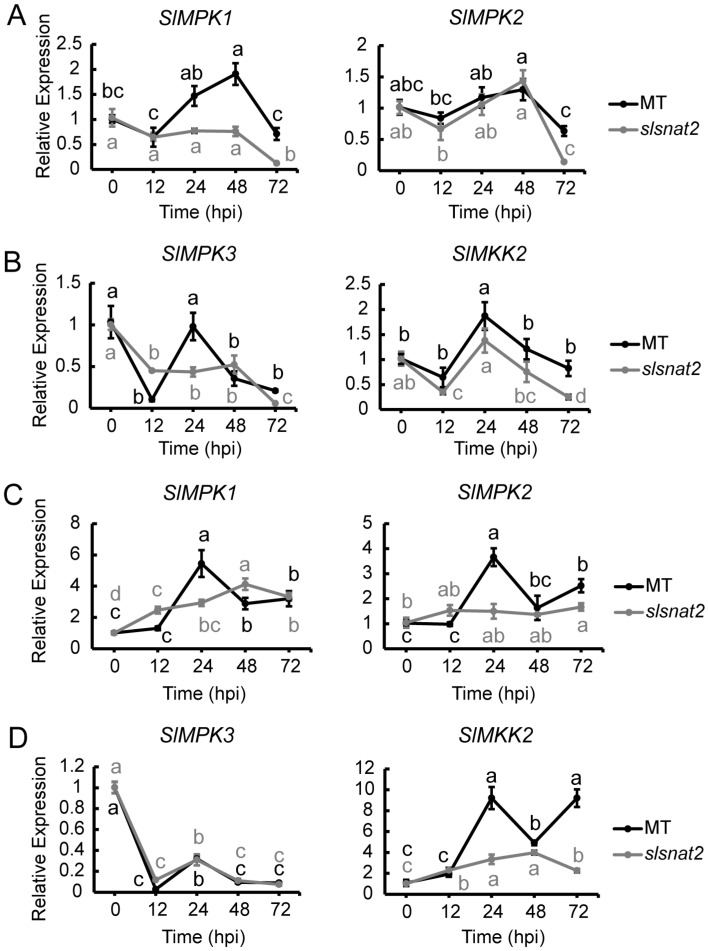
Expression analysis of MAPK defense pathway-related genes in roots and leaves of MT and *slsnat2* plants within 72 h post-inoculation with *R. solanacearum*. Root: (**A**) *SlMPK1* and *SlMPK2*. (**B**) *SlMPK3* and *SlMKK2*. Leaves: (**C**) *SlMPK1* and *SlMPK2*. (**D**) *SlMPK3* and *SlMKK2*. Relative transcript levels of MAPK defense pathway-related genes were examined in control and tomato mutant (*slsnat2*) plants at 0, 12, 24, 48, and 72 hpi by RT-qPCR, with *SlACTIN2* used as internal control. Data are represented as mean ± SEM (*n* = 3); Error bars: ±standard deviation with three biological replicates; Significant difference analysis was performed using LSD multiple comparisons and represented by letter labeling (*p* < 0.05). All experiments were repeated at least three times with similar results.

**Figure 8 ijms-26-06530-f008:**
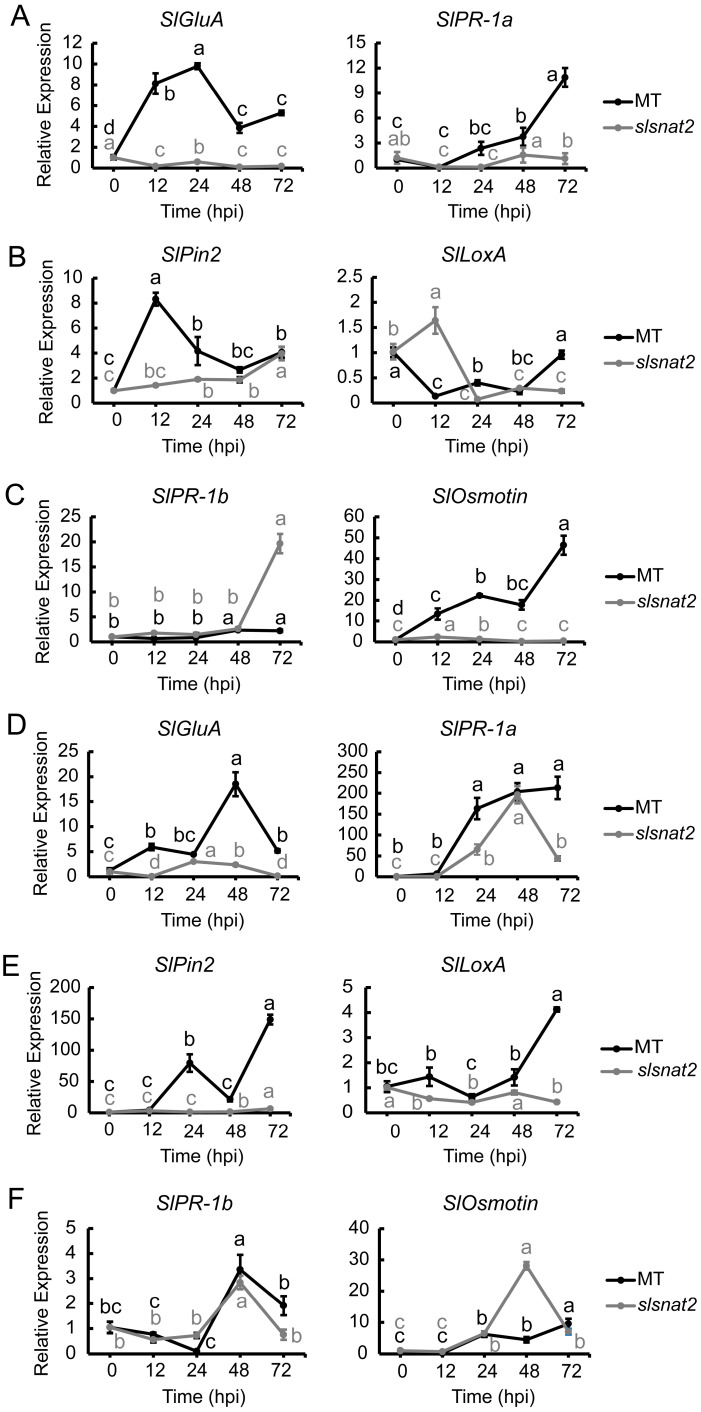
Expression analysis of SA, JA, and Eth signaling pathway-related genes in roots and leaves of MT and *slsnat2* plants within 72 h post-inoculation with *R. solanacearum*. Root, SA signaling pathway: (**A**) *SlGluA* and *SlPR-1a*. JA pathway: (**B**) *SlPin2* and *SlLoxA*. Eth signaling pathway: (**C**) *SlPR-1b* and *SlOsmotin*. Leaves, SA signaling pathway: (**D**) *SlGluA* and *SlPR-1a*. JA pathway: (**E**) *SlPin2* and *SlLoxA*. Eth signaling pathway: (**F**) *SlPR-1b* and *SlOsmotin*. Relative transcript levels of genes related to SA, JA, and Eth signaling pathways were examined in control and tomato mutant (*slsnat2*) plants at 0, 12, 24, 48, and 72 hpi by RT-qPCR, with *SlACTIN2* used as internal control. Data are represented as mean ± SEM (*n* = 3); Error bars: ±standard deviation with three biological replicates. Significant difference analysis was performed using LSD multiple comparisons and represented by letter labeling (*p* < 0.05). All experiments were repeated at least three times with similar results.

**Figure 9 ijms-26-06530-f009:**
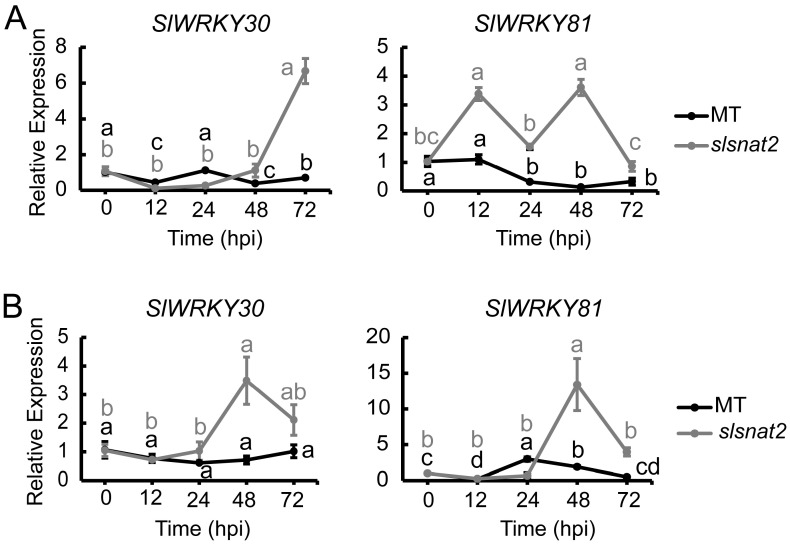
Expression analysis of *SlWRKY30* and *SlWRKY81* in roots (**A**) and leaves (**B**) of MT and *slsnat2* within 72 h post-inoculation with *R. solanacearum*. Relative transcript levels of *SlWRKY30* and *SlWRKY81* were examined in control and tomato mutant (*slsnat2*) plants at 0, 12, 24, 48, and 72 hpi by RT-qPCR, with *SlACTIN2* used as internal control. Data are represented as mean ± SEM (*n* = 3); Error bars: ±standard deviation with three biological replicates. Significant difference analysis was performed using LSD multiple comparisons and represented by letter labeling (*p* < 0.05). All experiments were repeated at least three times with similar results.

**Figure 10 ijms-26-06530-f010:**
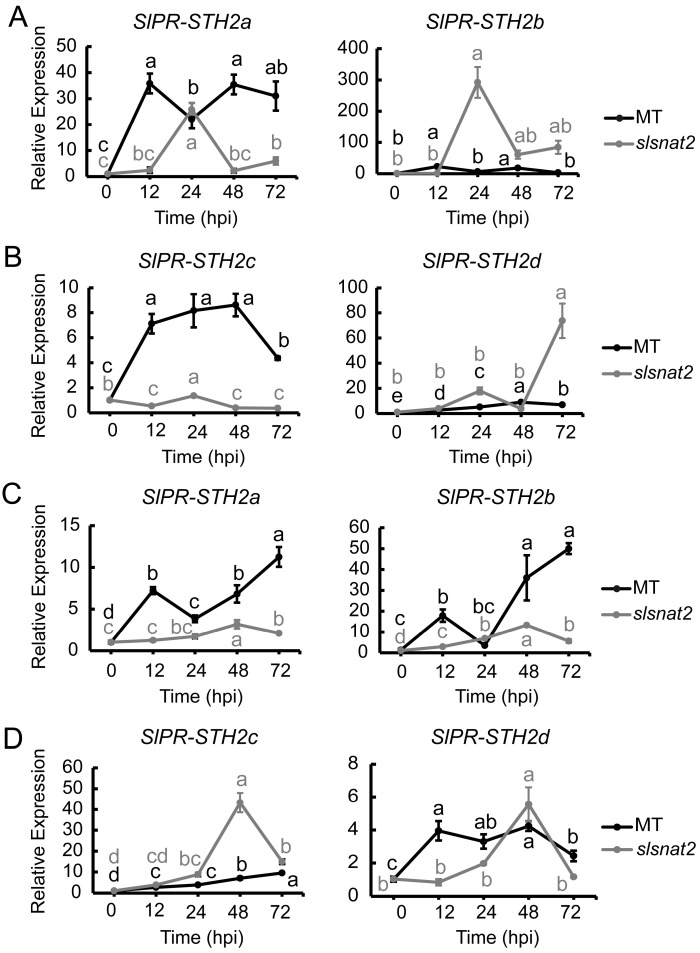
Expression analysis of *SlPR-STHs* in roots and leaves of MT and *slsnat2* within 72 h post-inoculated with *R. solanacearum*. Root: (**A**) *SlPR-STH2a* and *SlPR-STH2b*. (**B**) *SlPR-STH2c* and *SlPR-STH2d*. Leaves: (**C**) *SlPR-STH2a* and *SlPR-STH2b*. (**D**) *SlPR-STH2c* and *SlPR-STH2d*. Relative transcript levels of *SlPR-STHs* were examined in control and tomato mutant (*slsnat2*) plants at 0, 12, 24, 48, and 72 hpi by RT-qPCR, with *SlACTIN2* used as internal control. Data are represented as mean ± SEM (*n* = 3); Error bars: ±standard deviation with three biological replicates. Significant difference analysis was performed using LSD multiple comparisons and represented by letter labeling (*p* < 0.05). All experiments were repeated at least three times with similar results.

## Data Availability

All relevant data generated or analyzed are included in the manuscript and the Supporting Materials.

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
