# Peer review of "Serotonin N-acetyltransferase SlSNAT2 Positively Regulates Tomato Resistance Against Ralstonia solanacearum"

_ijms, 2025, doi:10.3390/ijms26136530_

Round 1
Reviewer 1 Report
Comments and Suggestions for Authors
Dear authors:
Bacterial wilt (BW) is a globally serious soil-borne disease in a wide range of plants, caused by diverse strains of Ralstonia solanacearum. This study indicates that SlSNAT2 enhances the resistance of tomatoes to tomato red blight pathogen by regulating reactive oxygen species homeostasis, callose deposition, MAPK signal transduction, hormone pathways and PR gene transcription, which is of great significance for the research of tomato disease resistance. The main question is why the SlSNAT2 overexpression line was not constructed in the text? It is necessary to determine the changes in the resistance level of SlSNAT2 overexpression strains to BW.
In addition, specific details are as follows:
- Why study the SlSNAT2 gene? From what information can this gene be obtained, for instance, is transcriptome sequencing induced by pathogenic bacteria or can a related gene be randomly selected from tomatoes for research?
- Line 35, The first appearance of Ralstonia solanacearumrequires full writing and abbreviation.
- Some of the text in Figure 2A is not fully displayed.
- Line 133, solanacearum needs to be in italics.
- The authors of the references were not listed completely. For example, for Hongbo Zhao, you only wrote Zhao H.
- In discussion. The discussion section could benefit from more explicit comparisons to similar international studies, highlighting the novelty and broader applicability of the findings.
Comments on the Quality of English Language
Reviewer 2 Report
Comments and Suggestions for Authors
This manuscript presents significant findings on the role of SISNAT2 in enhancing tomato resistance to Ralstonia solanacearum, providing insights into the intersection of melatonin signaling and biotic stress defense in crops. The integration of physiological and molecular approaches significantly enhances our understanding of melatonin-mediated immunity. However, revisions are necessary to enhance the manuscript, particularly in the introduction and methodology sections.
- The abstract is well-structured
- The introduction presented a rich background
- Lines: 34-38 avoid long sentences
- Line 42: “60-100%” is a wide range that needs to be contextually explained or narrowed
- Line 50: add an “and” after [11],
- No need “crop” in yield enhancement or improvement in photosynthetic efficiency
- Methodology section several gene/protein names are inconsistent (slsnat2 vs SISNAT2)
- Italicize gene names and use regular font for proteins
- The TTC medium composition, LC-MS/MS parameters need more details
- In results section italicize gene names
- The results section must be in the past tense
- Make sure the use of SISNAT2 is consistent in the result section
- Use a consistent name slsnat2 vs SISNAT2
- The discussion section is well structured
- Add a conclusion that summarizes the findings of your study and the perspectives
Good luck!!!
